# Loss of Frataxin activates the iron/sphingolipid/PDK1/Mef2 pathway in mammals

Kuchuan Chen[1†], Tammy Szu-Yu Ho[2†], Guang Lin[3], Kai Li Tan[1], Matthew N Rasband[1,2], Hugo J Bellen[1,2,3,4,5*]

[1]Program in Developmental Biology, Baylor College of Medicine, Houston, United States; [2]Department of Neuroscience, Baylor College of Medicine, Houston, United States; [3]Department of Molecular and Human Genetics, Baylor College of Medicine, Houston, United States; [4]Howard Hughes Medical Institute, Baylor College of Medicine, Houston, United States; [5]Jan and Dan Duncan Neurological Research Institute, Houston, United States

**Abstract** Friedreich's ataxia (FRDA) is an autosomal recessive neurodegenerative disease caused by mutations in *Frataxin* (*FXN*). Loss of *FXN* causes impaired mitochondrial function and iron homeostasis. An elevated production of reactive oxygen species (ROS) was previously proposed to contribute to the pathogenesis of FRDA. We recently showed that loss of *frataxin homolog* (*fh*), a *Drosophila* homolog of *FXN*, causes a ROS independent neurodegeneration in flies (Chen et al., 2016). In *fh* mutants, iron accumulation in the nervous system enhances the synthesis of sphingolipids, which in turn activates 3-phosphoinositide dependent protein kinase-1 (Pdk1) and myocyte enhancer factor-2 (Mef2) to trigger neurodegeneration of adult photoreceptors. Here, we show that loss of *Fxn* in the nervous system in mice also activates an iron/sphingolipid/PDK1/Mef2 pathway, indicating that the mechanism is evolutionarily conserved. Furthermore, sphingolipid levels and PDK1 activity are also increased in hearts of FRDA patients, suggesting that a similar pathway is affected in FRDA.

*For correspondence: hbellen@bcm.edu

†These authors contributed equally to this work

## Introduction

FRDA is the most prevalent form of recessive cerebellar ataxia characterized by neurodegeneration and cardiomyopathy. Patients with FRDA exhibit a progressive degeneration of the dorsal root ganglia, sensory peripheral nerves, and dentate nuclei of the cerebellum. Most individuals develop several neurological symptoms at a juvenile age, but individuals with adult or late onset are occasionally reported (*Koeppen, 2011*). Mutations in *FXN* were discovered as the primary cause of FRDA (*Campuzano et al., 1996*). Studies of the *yeast frataxin homolog* identified a role for *FXN* in iron-sulfur cluster biosynthesis and iron homeostasis (*Babcock et al., 1997*; *Mühlenhoff et al., 2002*; *Rötig et al., 1997*). Loss of *FXN* has been shown to cause mitochondrial dysfunction in several model organisms (*Anderson et al., 2005*; *Puccio et al., 2001*; *Rötig et al., 1997*). Yet, whether iron accumulates in the nervous system is still under debate, as it has been reported that iron levels are not altered in the nervous system of a conditional knockout mouse model and in FRDA patients (*Puccio et al., 2001*; *Simon et al., 2004*; *Solbach et al., 2014*). However, in FRDA patients, iron was shown to accumulate in the dentate nuclei or in glia cells of dorsal root ganglia (*Boddaert et al., 2007*; *Koeppen et al., 2009*). In sum, the iron deposition phenotype is still controversial in mouse models and FRDA patients, and the role of iron in the pathophysiology of FRDA has not yet been determined.

It has been proposed that ROS, generated by impaired mitochondrial electron transport chain complexes or iron deposition, mediate the pathogenesis in FRDA (*Bayot et al., 2011*; *Santos et al., 2010*). However, emerging evidence suggests that ROS are not elevated in several model organisms, and several clinical trials based on antioxidant therapy have shown no or limited benefit in FRDA patients (*Babcock et al., 1997*; *Llorens et al., 2007*; *Macevilly and Muller, 1997*; *Santhera Pharmaceuticals, 2010*; *Seznec et al., 2005*; *Shidara and Hollenbeck, 2010*). Hence, the pathogenesis of this disease is still poorly understood.

In a recent study, we described the first fly mutant allele of *frataxin homolog* (*fh*), the fly homolog of *FXN* (*Chen et al., 2016*). We showed that loss of *fh* in *Drosophila* leads to iron accumulation in the nervous system, which in turn induces sphingolipid synthesis and ectopically activates Pdk1 and Mef2. Mef2 activation triggers the aberrant transcription of downstream target genes that causes the degeneration of fly photoreceptors (*Chen et al., 2016*). However, whether this pathway is also affected in vertebrates upon loss of *FXN* is unknown. Here, we removed the endogenous mouse *Fxn* gene by using an adeno-associated virus (AAV) and CRISPR/Cas9 strategy (*Swiech et al., 2014*). We show that the iron/sphingolipid/PDK1/Mef2 pathway is activated in mice upon loss of *Fxn*. Furthermore, sphingolipids and PDK1 activity are also up-regulated in hearts of FRDA patients, suggesting that this pathway is also affected in FRDA. Hence, down-regulation of this pathway provides possible new targets to interfere with disease progression.

## Results

To test if the iron/sphingolipid/PDK1/Mef2 pathway is activated in vertebrates in vivo upon loss of *Fxn*, we reduced *Fxn* levels in the mouse nervous system by using the AAV and CRISPR/Cas9 system (*Swiech et al., 2014*). We designed two different *Fxn* single guide RNAs (sgRNAs) and tested their efficiency. The *Fxn*-sgRNAs induce lesions in the *Fxn* locus and reduce the levels of FXN protein when compared to the *LacZ*-sgRNA control in Neuro-2a cells (*Figure 1—figure supplement 1A and B*). To deliver *LacZ*- or *Fxn*-sgRNA into the nervous system, we injected AAV that carried sgRNA into the ventricle of Rosa26-Cas9 knock-in newborn mouse brains (*Platt et al., 2014*), and the efficiency of AAV infection into the cortical neurons was examined. As shown in *Figure 1—figure supplement 1C*, the majority of cortical neurons are infected with AAV, and the mRNA levels of *Fxn* in *Fxn*-sgRNA mice are less than 40% of the *LacZ*-sgRNA control mice (*Figure 1—figure supplement 1D*).

At approximately four months of age the *Fxn*-sgRNA mice (mice with *Fxn*-sgRNA AAV injection) develop an abnormal appearance when compared to *LacZ*-sgRNA control mice. These include a smaller body size and a hunchback phenotype at P130 (*Figure 1A*). Interestingly, these features are similar to the neuronal conditional knockout mice that were previously reported (*Puccio et al., 2001*). In addition, the *Fxn*-sgRNA mice become much less mobile than their *LacZ*-sgRNA littermates (*Videos 1* and *2*). The *Fxn*-sgRNA mice also display an aberrant reflex when lifted by the tail as they fail to spread their hind limbs (*Figure 1B*). To evaluate the sensorimotor coordination, we performed a rotarod test at two different ages. At P50, we observe no difference between *LacZ*-sgRNA and *Fxn*-sgRNA mice. However, *Fxn*-sgRNA mice show a significant reduction of latency to fall from the rod at age of P130 (*Figure 1C*). Besides the deficit on the rotarod test, *Fxn*-sgRNA mice also show a severe impairment in the wire hang test at P130 (*Figure 1D*). In summary, a removal of *Fxn* in a substantial neuronal population in the newborn mouse brain causes a severe impairment of neuronal function.

We further characterized the neuronal phenotype by examining brain sections at P130. We observed that the activating transcription factor 3 (ATF3), a marker that is typically associated with cellular stress response and neuronal damage (*Tsujino et al., 2000*), is up-regulated in cortical neurons of *Fxn*-sgRNA mice (*Figure 2A*). However, no ATF3 positive cells were observed in *LacZ*-sgRNA mice (*Figure 2A*). In addition, the axon initial segments, specialized membrane regions where $Na^+$ channels cluster to initiate action potentials (*Szu-Yu Ho and Rasband, 2011*), are lost in cortical neurons of *Fxn*-sgRNA mice. As shown in *Figure 2B*, the scaffolding protein required for the formation and maintenance of the axon initial segment, ankyrin-G (AnkG), is severely reduced or lost in cortical neurons in *Fxn*-sgRNA mice, especially in ATF3 positive neurons (*Figure 2B*). These data suggest that loss of *Fxn* in cortical neurons leads to severe neuronal injury and damage.

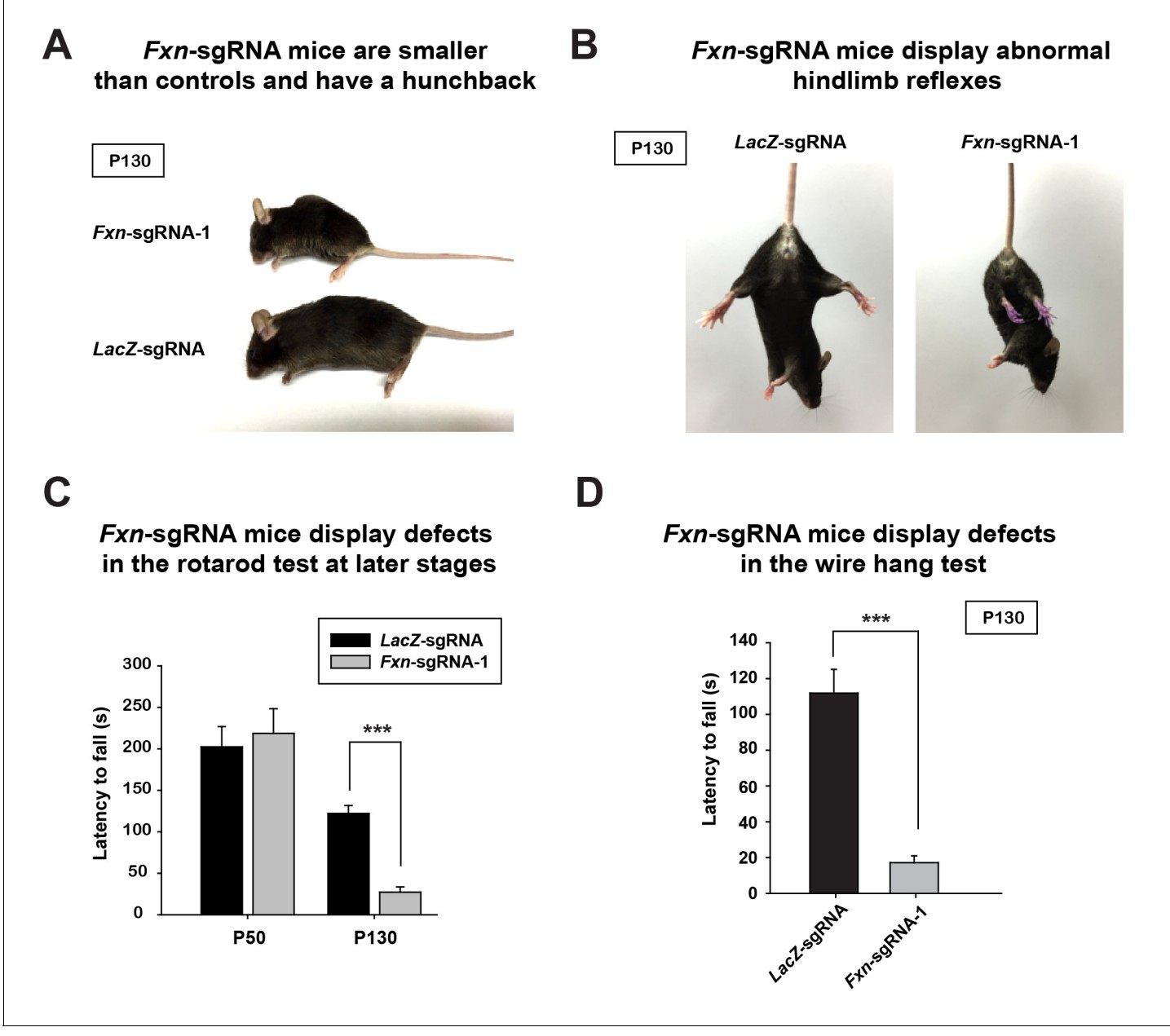

**Figure 1.** Phenotypic characterization of *Fxn*-sgRNA mice. (**A**) Body size of *LacZ*- and *Fxn*-sgRNA mice at P130. *Fxn*-sgRNA mice exhibit a hunchback phenotype. (**B**) Hindlimb reflexes of *LacZ*- and *Fxn*-sgRNA mice at P130. (**C**) Rotarod test of *LacZ*- and *Fxn*-sgRNA mice at P50 (n = 5) and P130 (n = 6). (**D**) Wire hang test of *LacZ*- and *Fxn*-sgRNA mice at P130. n = 6. Data are presented as mean ± SEM. ***, p<0.001, Student's t-test.

The following figure supplement is available for figure 1:

**Figure supplement 1.** Removal of *Fxn* using AAV and CRISPR/Cas9.

To assess if the iron/sphingolipid/PDK1/Mef2 pathway is activated in the brains of *Fxn*-sgRNA mice, we assessed if iron levels are increased upon loss of *Fxn*. To examine $Fe^{3+}$ levels, we performed Perls' staining and used 3,3'-diaminobenzidine (DAB) to enhance the signal. At P130 the DAB signal is strongly increased in brain sections of *Fxn*-sgRNA mice when compared to control mice (*Figure 3A*). To further characterize the iron distribution, we co-stain iron with neuronal and glia markers in the same brain sections. Since strong DAB signal interferes with the fluorescence, we

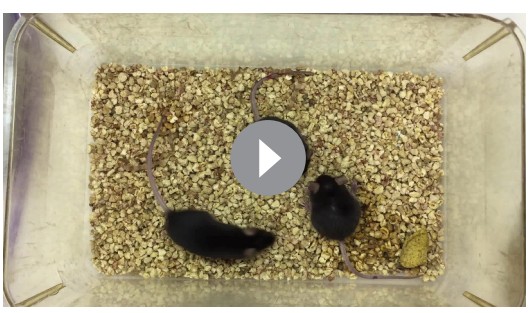 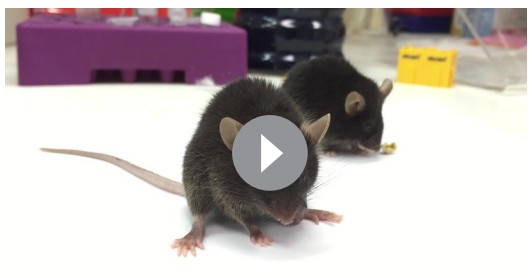

**Video 1.** *LacZ-* and *Fxn*-sgRNA mice at P130.

**Video 2.** *Fxn*-sgRNA mice at P130.

chose the cortex region with relatively weak DAB deposits of *Fxn*-sgRNA mice. Interestingly, the majority of DAB signals are co-localized with a neuronal marker but not glial markers, suggesting that $Fe^{3+}$ accumulates in neurons, although there are some weak diffuse signals distributed outside of the cell body (*Figure 3* and *Figure 3—figure supplement 1*). These data show that $Fe^{3+}$ accumulates in the nervous system upon loss of *Fxn*. The fact that we observed an obvious $Fe^{3+}$ accumulation, whereas previous reports did not, is likely due to the use of DAB to enhance the Perls' Prussian blue staining, as we did not observe any signal with Perls' Prussian blue staining alone (*Figure 3—figure supplement 2A*).

We next assessed $Fe^{2+}$ levels by using RPA (*Petrat et al., 2002*). RPA is a cell-permeable fluorescent dye that selectively accumulates in mitochondria, and its fluorescence is stoichiometrically quenched by $Fe^{2+}$ (*Petrat et al., 2002*). Interestingly, we observed $Fe^{2+}$ accumulation in the brain of *Fxn*-sgRNA mice at P60, a time point at which we did not observe behavioral defects in the *Fxn*-sgRNA mice. As shown in *Figure 3B*, the RPA fluorescence is quenched in cells of cortical layer IV and V of *Fxn*-sgRNA mice when compared to *LacZ*-sgRNA controls, indicating an accumulation of $Fe^{2+}$ in the nervous system upon loss of *Fxn*. As a negative control, we used RPAC, a dye that is structurally very similar to RPA but is insensitive to $Fe^{2+}$ mediated fluorescence quenching. RPAC is present at similar levels in both *Fxn*-sgRNA and *LacZ*-sgRNA brains (*Figure 3B*), suggesting that the decreased fluorescence of RPA in *Fxn*-sgRNA mice is not due to impaired dye uptake but due to $Fe^{2+}$ quenching.

It was previously proposed that iron deposition triggers free radical production via Fenton's reaction. We therefore measured oxidative stress by examining the levels of 4-hydroxynonenal (4-HNE), a major product of lipid peroxidation (*Ayala et al., 2014*). However, we did not observe an increase of 4-HNE in *Fxn*-sgRNA brain tissues (*Figure 3—figure supplement 2B*), consistent with previous observations in flies (*Chen et al., 2016*). We next determined if iron triggers the activation of the PDK1/Mef2 pathway in *Fxn*-sgRNA mice. Phosphorylation of S241 in the PDK1 activation loop is required for its activity (*Casamayor et al., 1999*). We observed an increase in phosphorylation levels of S241 in *Fxn*-sgRNA mice compared to the *LacZ*-sgRNA mice at P60, prior to the onset of symptoms (*Figure 3C* and *Figure 3—figure supplement 2C*). To assess if Mef2 is activated in *Fxn*-sgRNA mice, we determined the mRNA levels of several validated Mef2 downstream targets (*Anderson et al., 2004*; *Ewen et al., 2011*; *Potthoff et al., 2007*; *Woronicz et al., 1995*). As shown in *Figure 3D*, many, but not all, transcripts are up-regulated 1.5–3 fold in *Fxn*-sgRNA mice at P60. This is consistent with previous findings that only about half of Mef2 targets are up-regulated (to 2–3 fold) when Mef2A is overexpressed (*Ewen et al., 2011*). In summary, these results provide evidence that the PDK1/Mef2 pathway is activated in the vertebrate nervous system upon loss of *Fxn* prior to the onset of phenotypes that are observed about two months later.

Last, we tested if the same pathway is also activated in FRDA patients. As we were unable to obtain cerebella and dorsal root ganglia of patients, we used heart samples that were previously shown to accumulate $Fe^{3+}$ (*Bradley et al., 2000*; *Koeppen, 2011*). We first investigated the sphingolipid levels in frozen heart tissues of five controls and six FRDA patients. As shown in *Figure 4A*, a substantial fraction of the sphingolipids are increased in FRDA patients when compared to controls, including short chain ceramides (C14-C18), different dihydroceramides, dihydrosphingosines,

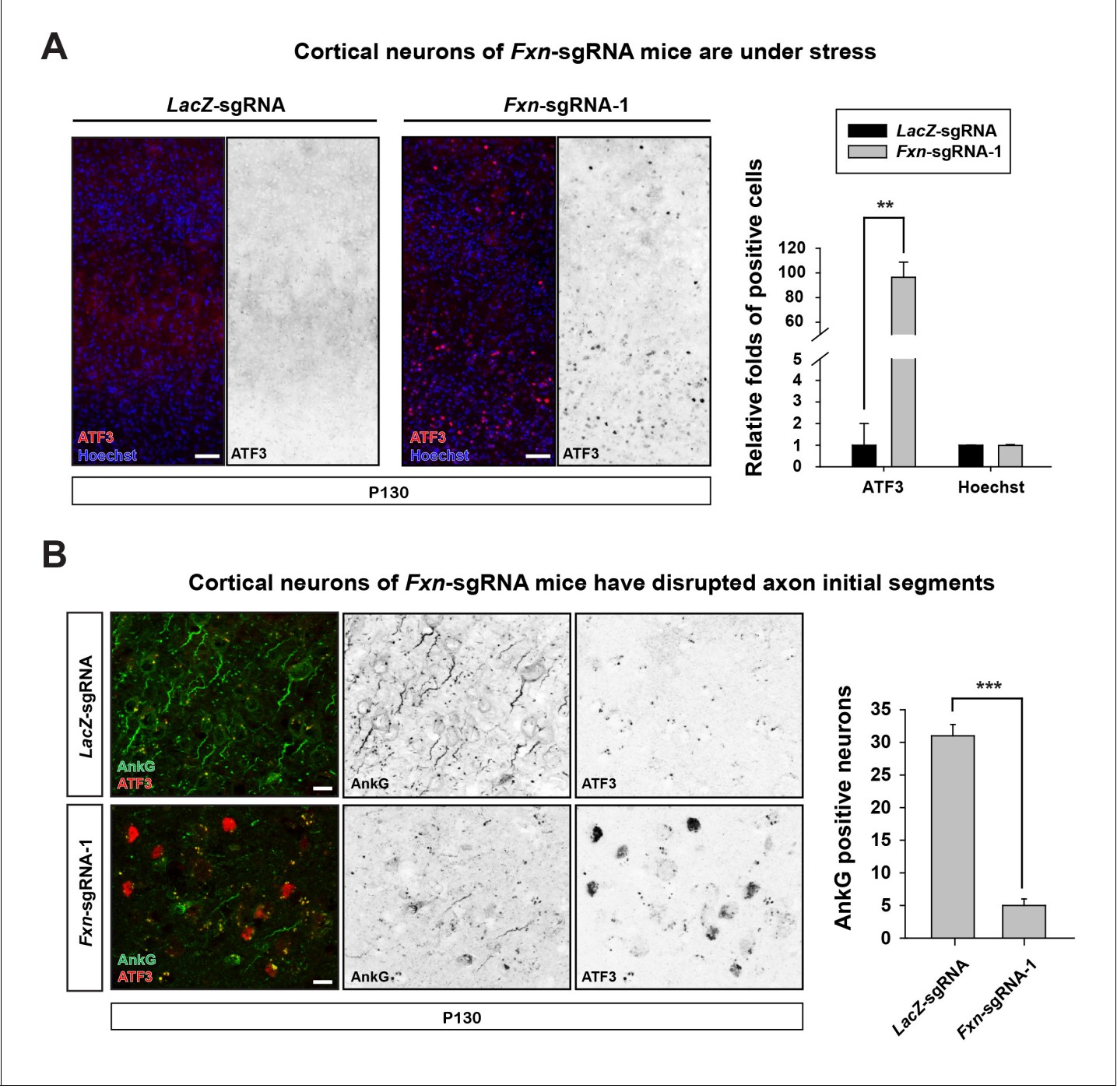

**Figure 2.** Loss of *Fxn* leads to neuronal insults. (A) Immunostaining of cerebral cortex layer I (top) to VI (bottom) of *LacZ*- and *Fxn*-sgRNA mice at P130. ATF3 is labeled in red, and nuclei are marked by Hoechst in blue. n = 3. Quantification is on the right. Scale bar, 50 μm. (B) Immunostaining of cortical neurons of *LacZ*- and *Fxn*-sgRNA mice at P130. Axon initial segments are marked by anti-AnkG antibody (green), and ATF3 is marked in red. n = 3. Quantification is on the right. Scale bar, 10 μm. Data are presented as mean ± SEM. **, p<0.01, ***, p<0.001, Student's t-test.

dihydrosphingosine-1-phosphate, and sphingosine. Finally, we tested if PDK1 activity is also increased in FRDA patients. The phosphorylation levels of S241 in PDK1 are elevated in FRDA patients when compared to controls (*Figure 4B*). Note that patient six showed no obvious increase in PDK1 phosphorylation when compared to the other patients. This patient had relatively mild

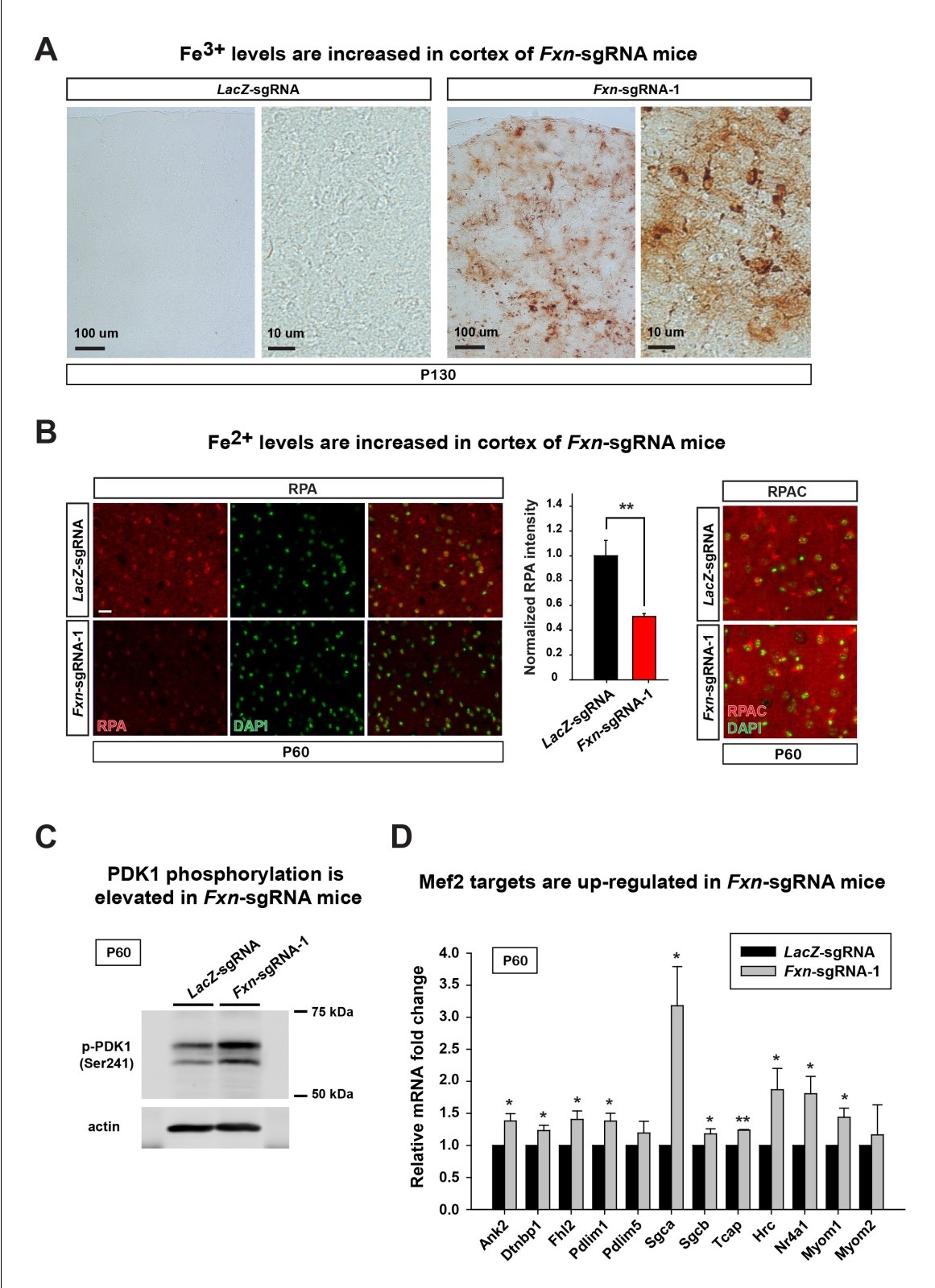

**Figure 3.** Loss of *Fxn* leads to an activation of the iron/sphingolipid/PDK1/Mef2 pathway.  (**A**) Perls' blue staining with DAB enhancement of the cerebral cortex layer I (top) to VI (bottom) of *LacZ*- and *Fxn*-sgRNA mice at P130. n = 3. Scale bar, 100 and 10 μm. (**B**) RPA and RPAC staining of mouse cortical neurons of *LacZ*- and *Fxn*-sgRNA mice at P60. RPA and RPAC fluorescence is in red, and nuclei are labeled by DAPI in green. n = 3. Scale bar:
*Figure 3 continued on next page*

*Figure 3 continued*

20 µm. (**C**) Immunoblot of PDK1 phosphorylation levels of brains of *LacZ*- and *Fxn*-sgRNA mice at P60. (**D**) mRNA levels of Mef2 downstream targets of *LacZ*- and *Fxn*-sgRNA mice at P60. n = 3. Data are presented as mean ± SEM. *, p<0.05. **, p<0.01, Student's t-test.

The following figure supplements are available for figure 3:

**Figure supplement 1.** $Fe^{3+}$ distribution in brains of *LacZ*- and *Fxn*-sgRNA mice.

**Figure supplement 2.** $Fe^{3+}$ staining and immunoblot of *LacZ*- and *Fxn*-sgRNA mice.

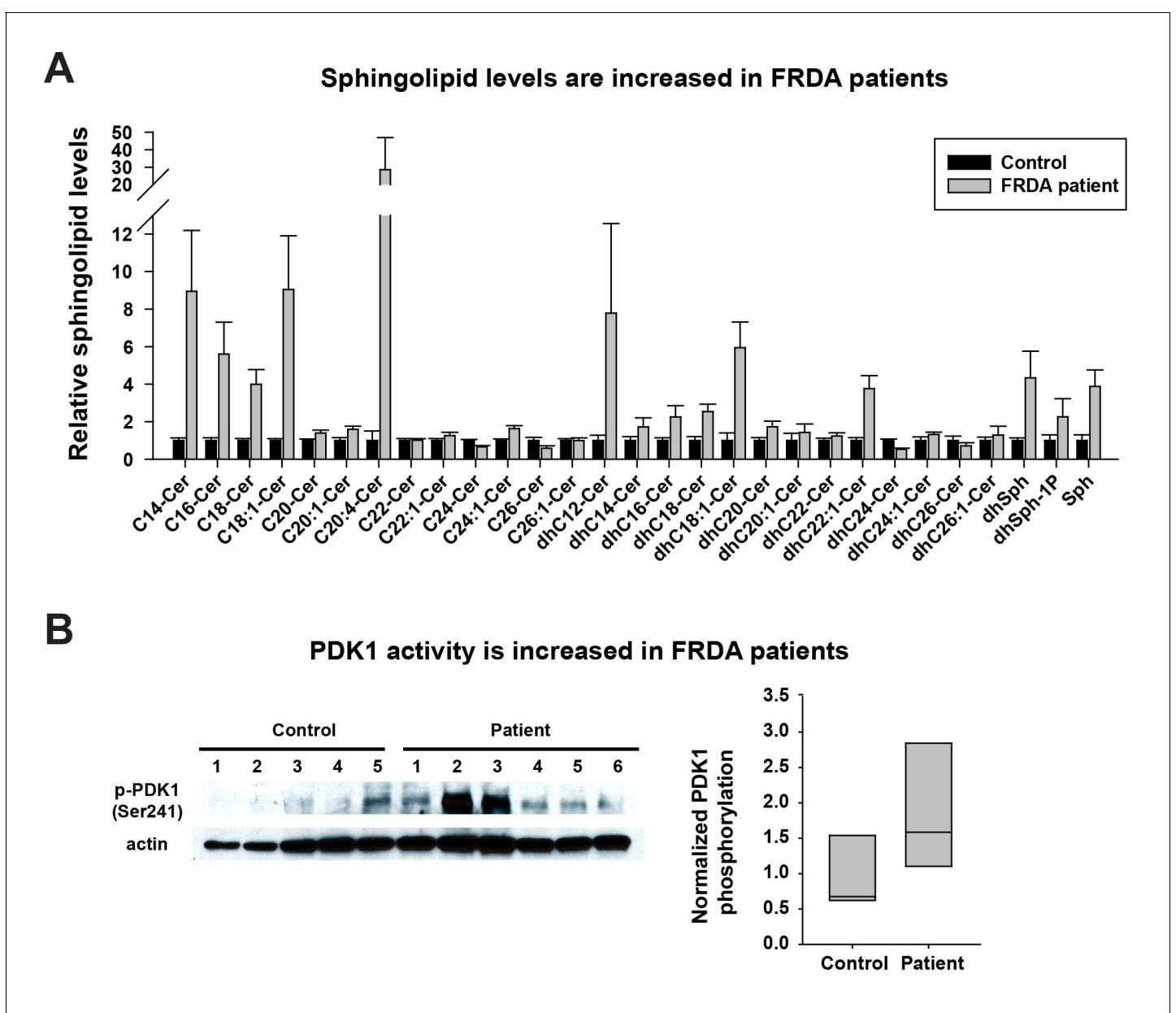

**Figure 4.** Sphingolipid levels and PDK1 activity are increased in heart tissues of FRDA patients. (**A**) Sphingolipid profiling of heart tissues from controls (n = 5) and FRDA patients (n = 6). Data are presented as mean ± SEM. (**B**) Immunoblot of PDK1 phosphorylation levels of heart tissues from controls (n = 5) and FRDA patients (n = 6). Box plot of PDK1 phosphorylation is on the right.

symptoms and a later onset of disease (age 18) than other patients (ages 4 to 9). The age of death was also much later in patient 6 (age 67) compared to the other patients (age 25–37). Similarly, the increased levels of sphingolipids in patient six are also less pronounced than other patients. In sum, these data suggest that the iron/sphingolipid/PDK1/Mef2 pathway is activated in FRDA patients and that it may contribute to the pathogenesis of FRDA.

## Discussion

Here we show that removal of *Fxn* in the central nervous system using AAV and CRISPR/Cas9 causes behavioral and neurological phenotypes in mice. This new FXN mouse model exhibits several similar phenotypes to the previous neuronal conditional knockout mouse model, including smaller body size, hunchback phenotype, impaired locomotion, and shortened life span (*Puccio et al., 2001*). By using RPA and Perls'/DAB stainings, more sensitive iron staining assays than Perls' Prussian blue staining alone, we found that $Fe^{2+}$ and $Fe^{3+}$ levels are increased in the cerebral cortex of *Fxn*-sgRNA mice, and the activities of PDK1 and Mef2 are up-regulated, suggesting that the iron/sphingolipid/PDK1/Mef2 pathway, previously identified in fly *fh* mutants, is also activated in *Fxn*-sgRNA mice. Importantly, sphingolipids and PDK1 activity are also up-regulated in the hearts of FRDA patients, indicating that a similar pathway is also activated and may contribute to the pathogenic mechanism of FRDA.

Iron toxicity has been long considered as a possible pathogenic contributor of FRDA. However, the iron accumulation phenotypes have never been reported in the nervous system in an inducible conditional knockout mouse model (*Martelli and Puccio, 2014*; *Puccio et al., 2001*; *Simon et al., 2004*). We provide the first direct evidence that both $Fe^{2+}$ and $Fe^{3+}$ accumulate in the brain upon loss of *Fxn*. In addition, we noticed that irons are not only deposited inside the neuronal cell body but also present in the extracellular space. For example, DAB staining showed a diffuse pattern in and surrounding the neuronal cell body (*Figure 3A* and *Figure 3—figure supplement 1*), and the RPA fluorescence is similarly reduced in the extracellular space (*Figure 3B*), suggesting that iron accumulation occurs both intracellularly and extracellularly.

It is worth noting that the iron/sphingolipid/PDK1/Mef2 pathway is conserved from yeast to fly (*Chen et al., 2016*; *Lee et al., 2012*), and that this pathway is also activated in *Fxn*-sgRNA mice and the hearts of FRDA patients. Interestingly, genetic variations at the 5'UTR of Mef2C that promote Mef2C transcription are associated with hypertrophic cardiomyopathy (*Alonso-Montes et al., 2012*). In addition, overexpression of Mef2A or Mef2C causes cell sarcomeric disorganization and elongation in cultured cardiomyocytes (*Xu et al., 2006*). These data suggest that an activation of iron/sphingolipid/PDK1/Mef2 pathway may play a role in the induction of cardiomyopathy in FRDA patients. Inhibition of this pathway may not only suppress neurodegeneration, but also prevent or delay cardiomyopathy, the major cause of lethality in FRDA. In sum, this pathway may provide us with therapeutic targets based on antisense oligonucleotide strategies (*Ackermann et al., 2012*), an option that should be explored.

## Materials and methods

### sgRNA design

sgRNAs were designed using the CRISPR Design Tool (http://crispr.mit.edu). Control sgRNA sequence was designed to target *lacZ* gene from *Escherichia coli* (*Swiech et al., 2014*): TGCGAA TACGCCCACGCGAT. *Fxn*-sgRNA sequences targeting the 5' end of the *Fxn* gene locus were used: *Fxn* sgRNA-1, AGGGAACCGATCGTAACCTG and *Fxn* sgRNA-2, TTCGGAGGTCGCGCAGCCGT. sgRNAs DNA duplex were subcloned into the AAV-SpGuide vector (Addgene) (*Swiech et al., 2014*). To test sgRNA efficiency in genome editing, AAV-SpCas9 (Addgene) and AAV-sgRNA constructs were co-transfected into Neuro-2a cells. Cells were harvested 48 hr after transfection and the genomic DNA was extracted. A ~1 Kb fragment containing the *Fxn* sgRNA sequence was PCR amplified with the following primers, Fxn_F: 5'-CACCAGCGATCTTCAGAATCACC-3' and Fxn-R: 5'-GGAGGCAGGAGGATCACAGG-3'. PCR products were then re-hybridized and cleaved by T7 endonuclease (NEB). Fragmented PCR products were then analyzed by 2% agarose gel. To test sgRNA efficiency in Fxn protein levels, AAV-SpCas9 (Addgene) and AAV-sgRNA constructs were co-

transfected into Neuro-2a cells. As anti-FXN antibody cannot recognize the endogenous FXN proteins, we co-transfected *Fxn*-cDNA construct into Neuro-2a cells to allow FXN overexpression. Cells were then harvested and the FXN protein levels were analyzed by immunoblotting.

## AAV production and purification

AAV vectors were produced at the Gene Vector Core, Baylor College of Medicine. Dulbecco's modified Eagle's medium (DMEM) with high glucose (4.5 g/l) and antibiotic-antimycotic were purchased from GenDEPOT (Katy, TX), and fetal bovine serum (FBS) was purchased from Sigma (St. Louis, MO). HEK293T cells were grown in DMEM supplemented with 10% FBS and 1x antibiotic-antimycotic. DJ/8 Rep-Cap plasmid (*Grimm et al., 2008*) was purchased from Cell Biolabs (San Diego, CA). AAVs were packaged by three plasmids transfection (transfer vector, DJ/8, and helper plasmid) using iMFectin Poly DNA Transfection Reagent (GenDEPOT). In brief, 80–90% confluent HEK293T cells in a 15 cm dish were split into 6×15 cm dish one day before transfection. Next day, the media was removed and replaced with 9 ml of fresh DMEM/5% FBS. 24 µl of iMFectin reagent were diluted to 0.5 ml by DMEM and then added to 0.5 ml of DMEM containing 4 µg of helper plasmid, 2 µg of DJ/8 and 2 µg of transfer vector. The transfection mixture was vortexed for 10 s and incubated for 15 min at room temperature before overlay on HEK293T cells without removing media. After 4 hr, 10 ml of DMEM/5% FBS were added and cells were incubated in humidified $CO_2$ incubator. AAV purification was performed by the method developed by Ayuso et al (*Ayuso et al., 2010*) with modifications. Three days after transfection, cells were collected by centrifugation at 2500 x g for 10 min. Cell culture supernatant was retained for subsequent precipitation. The cell pellet was re-suspended in 1 ml of 50 mM Tris pH8.0 containing 5 mM $MgCl_2$ and 0.15 M NaCl per plate, lysed by adding 0.1 vol of 5% sodium deoxycholate at room temperature for 30 min and then incubated with 10 µg/ml of DNase I and RNase A for 1 hr at 37°C. Cell lysates were clarified by centrifugation at 5000 x g for 10 min at 4°C. Cell culture supernatant was incubated with 10 µg/ml of DNase I and RNase A for 1 hr at 37°C and then incubated at 4°C overnight after addition of a stock solution of 40% polyethylene glycol (PEG) 8000 containing 2.5 M NaCl to final concentration of 8% PEG. AAV in cell culture supernatant was collected by centrifugation at 2500 x g for 30 min. The pellets containing AAV were re-suspended in a minimal volume of HBS (50 mM HEPES, 0.15 M NaCl, 1% Sarcosyl, 20 mM EDTA, pH8.0). Cell associated and secreted AAVs were combined for the subsequent iodixanol density centrifugation (*Zolotukhin et al., 1999*). AAVs were dialyzed against phosphate buffered saline without $Mg^{2+}$ and $Ca^{2+}$ and the titer was determined by real time PCR using vector specific primers (*Ljungberg et al., 2012*).

## AAV injection into the neonatal mouse brain

Rosa26-Cas9 knockin mice (*Gt(ROSA)26Sor*[tm1.1(CAG-cas9,-EGFP)Fez h]*/J*, Stock No. 024858, the Jackson Laboratory, gift from Huda Zoghbi) were used for AAV injection. P1 neonates were cryo-anesthesized at 0°C for 3 min before injection. AAVDJ-8 carrying *Fxn*-sgRNA or control *LacZ*-sgRNA ($2 \times 10^{14}$ gc/ml) was injected into the ventricles using a pulled glass needle. After injection, pups were laid on a warm pad until recovery. The brains were dissected at P60 and P130 after virus injection. All mouse work and procedures were approved by the Animal Care and Use Committee at Baylor College of Medicine and were performed in accordance with the guidelines of the U.S. National Institutes of Health (animal protocol: an-4634).

## Behavior test

*LacZ*- and *Fxn*-sgRNA mice were subjected to accelerating rotarod and wire hang tests as previously described (*Zhang et al., 2013*).

## Iron staining

To detect $Fe^{2+}$ levels of AAV injected mouse brain, the brains were dissected and embedded into 2.5% low-melting agarose (Sigma-Aldrich). The 100 µm brain sections were acquired by a vibratome (1000 Classic; Warner Instruments) and transferred into 24 well plate with ice-cold artificial cerebrospinal fluid (ACSF) (125 mM NaCl, 2.5 mM KCl, 1 mM $MgCl_2$ $6H_2O$, 1.25 mM $NaH_2PO_4$, 2 mM $CaCl_2.2H_2O$, 25 mM $NaHCO_3$, 25 mM Glucose, pH 7.4). The brain sections were then incubated with 10 µM RPA or RPAC (Squarix Biotechnology) in ACSF for 20 min at room temperature in the

dark. The samples were washed subsequently three times by ACSF. To detect $Fe^{3+}$ levels, brains were dissected and fixed in 4% paraformaldehyde for 1 hr on ice followed by immersion in 20% sucrose overnight at 4°C. Brains were then sectioned using a freezing stage Microtome (Microm KS 34, Thermo Scientific) and spread out on glass coverslips. Brain slices were then incubated with Perls' solution (1% $K_4Fe(CN)_6$ and 1% HCl in 0.4% Triton-PBS) for 10 min. After three quick washes in 0.4% Triton-PBS, samples were incubated for 5 min with DAB solution (10 mg DAB with 0.07% $H_2O_2$ in 0.4% Triton-PBS) to enhance the signal. The brains were then quickly washed three times with 0.4% Triton-PBS and mounted. The images were obtained with an Axio-imager Z1 microscope (Carl Zeiss) fitted with an AxioCam digital camera (Carl Zeiss). The images were analyzed by ZEN 2012 (Carl Zeiss).

## Immunofluorescence staining

For mouse brain staining, brains were dissected and fixed in 4% paraformaldehyde for 1 hr on ice followed by immersion in 20% sucrose overnight at 4°C. Brains were then sectioned using a freezing stage Microtome (Microm KS 34, Thermo Scientific) and spread out on glass coverslips for immunos-taining. The antibodies were used at the following concentrations: mouse anti-AnkG (N106/36, N106/20) (UC Davis/NIH), 1:200; rabbit anti-ATF3 (Santa Cruz) (RRID:AB_2058590), 1:500; chicken anti-GFP (ab13970, abcam) (RRID:AB_300798), 1:1000; rabbit anti-Iba1 (019–19741, Wako), 1:250; chicken anti-GFAP (ab4674, abcam) (RRID:AB_304558), 1:500; NeuroTrace (Thermo Scientific), 1:200; Alexa 488-, Cy3-, or Cy5-conjugated secondary antibodies (Jackson ImmunoResearch), 1:250. Samples were then mounted in Vectashield (Vector Laboratories) before being analyzed under a con-focal microscope. The images were obtained with an Axio-imager Z1 microscope (Carl Zeiss) fitted with an AxioCam digital camera (Carl Zeiss). The images were analyzed by ZEN 2012 (Carl Zeiss).

## Immunoblot

For the Neuro-2a cells (RRID:CVCL_0470), cells were harvested and lysed in ice-cold RIPA buffer (50 mM Tris-HCl, 150 mM NaCl, 1% NP-40, 1% sodium deoxycholate, 0.1% SDS, 50 mM NaF, 1 mM $Na_3VO_4$, 10% Glycerol, protease inhibitor Cocktail (Roche)) 48 hr after transfection. The samples were then centrifuged at 16,100 ×g for 10 min to remove nuclei and debris. For the brain tissues, mouse brains were dissected, frozen on dry ice and homogenized as previously described (*Ho et al., 2014*). Briefly, brains were homogenized in ice-cold homogenization buffer (0.32 M sucrose, 5 mM sodium phosphate, pH 7.2, 1 mM NaF, 1 mM $Na_3VO_4$, and protease inhibitors). The homogenates were centrifuged at 700×g for 10 min at 4°C to remove nuclei and debris. The supernatant was then centrifuged at 27,200×g for 90 min at 4°C, and the pellet was resuspended in homogenization buffer (3 ml/g of brain). Protein concentrations were measured using a Bradford assay (Bio-Rad). The samples were resolved by SDS-PAGE, transferred to nitrocellulose membrane, and immunoblotted with the following primary antibodies: rabbit anti-PDK1 (Cel Signaling) (RRID:AB_2236832); rabbit anti-p-PDK1 (Cell Signaling) (RRID:AB_2161134); mouse anti-actin (ICN Biomedicals) (RRID:AB_2336056); rabbit anti-FXN (Santa Cruz) (RRID:AB_2110677); rabbit anti-4-HNE (abcam). Secondary antibodies were from Rockland Immunochemicals. Images were acquired and analyzed with Odyssey CLx Infrared Imaging System (LI-COR Biotechnology).

## Mass spectrometry

Sphingolipid profiling of human heart tissues was performed by the Lipidomics Core at the Medical University of South Carolina as previously described (*Chen et al., 2016*).

## Reverse transcription–quantitative polymerase chain reaction (PCR)

Total RNA from mouse brains were extracted by Trizol RNA Isolation Reagents (Thermo Fisher Sci-entific, Sugar Land, Texas) following the manufacturer's instructions. The cDNA was synthesized by High-Capacity cDNA Reverse Transcription Kit (Applied Biosystems). Real-time PCR was performed using iQ SYBR Green Supermix (Bio-Rad) in a thermal cycler (iCycler; Bio-Rad Laboratories). The data were collected and analyzed using the optical module (iQ5; Bio-Rad Laboratories). The follow-ing primer pairs are used (5'to 3'): Gapdh forward (control primer), AGGTCGGTGTGAACGGA TTTG; Gapdh reverse (control primer), TGTAGACCATGTAGTTGAGGTCA; Ank2 forward, AGA TTACTGTGCAGCATAACAGG; Fxn forward, GCTGGAGGGAACCGATCGTA; Fxn reverse, TTCC

**Table 1.** Clinical and genetic data of 6 FRDA patients.

| No. | Gender | Age of onset | Age of death | GAA1 | GAA2 |
|-----|--------|--------------|--------------|------|------|
| 1 | M | 8 | 27 | 700 | 1070 |
| 2 | M | 4 | 37 | 674 | 674 |
| 3 | F | 5 | 25 | 800 | 1100 |
| 4 | F | 9 | 26 | 690 | 850 |
| 5 | M | 7 | 35 | 750 | 1100 |
| 6 | F | 18 | 67 | 621 | 766 |

The sizes for two GAA trinucleotide repeat expansions in the first intron of *FXN* in FRDA patients are shown with the smaller expansion designated GAA1 and the larger expansion designated GAA2.

TCAAATGCACCACGCAG; Ank2 reverse, TGGTTGTAAAGGAAACACACTCA; Dtnbp1 forward, TTAGCAGGTATGAGGATGCGT; Dtnbp1 reverse, GGTGCAGCAAATGGTTCTCTAC; Fhl2 forward, ATGACTGAACGCTTTGACTGC; Fhl2 reverse, CGATGGGTGTTCCACACTCC; Pdlim1 forward, TCGATGGGGAAGATACCAGCA; Pdlim1 reverse, TCTGTTCAGACCTGGATACTGTG; Pdlim5 forward, AGCAGCAAAATGGTAACCCTG; Pdlim5 reverse, CGTGAAGGGGTATCTTTTCCTTT; Sgca forward, GCAGCAGTAACTTGGATACCTC, Sgca reverse, AAAGGATGCACAAACACACGA; Sgcb forward, AGCACAACAGCAATTTCAAAGC, Sgcb reverse, AGGAGGACGATCACGCAGAT; Tcap forward, GATGCGCCTGGGTATCCTC; Tcap reverse, GATCGAGACAGGGTACGGC; Hrc forward, CAACCGATGTACCGAATGTGA; Hrc reverse, GGTAGCAGAATTGACAGTGCTG; Nr4a1 forward, TTGAGTTCGGCAAGCCTACC, Nr4a1 reverse, GTGTACCCGTCCATGAAGGTG; Myom1 forward, GCACGACCATGAGCCACTAC; Myom1 reverse, ACCCTTGAGAATGCCGGGA; Myom2 forward, AAAAGACACAAGCACTTTGACCA; Myom2 reverse, TGGGAGGATGACTGGGTGG.

## Control and FRDA patient heart specimens

Frozen heart specimens of FRDA patients were obtained from Dr. Arnulf Koeppen, Veterans Affairs (VA) Medical Center, Albany, NY, USA. Our receipt and analyses of these postmortem tissues was conducted with the approval of the Institutional Review Board at Baylor College of Medicine (protocol H-32191). *Table 1* shows clinical and genetic data of six patients with FRDA. Five frozen normal control heart specimens were obtained from National Disease Research Interchange, Philadelphia, PA, USA.

## Acknowledgements

We thank Karen Schulze for comments, Dr Huda Zoghbi for the Rosa26-Cas9 mice, and Dr Arnulf Koeppen for providing FRDA patient heart tissues. We thank National Disease Research Interchange for providing control heart tissues. We thank the Gene Vector Core at Baylor College of Medicine for the AAV production. We thank the Lipidomics Core at the Medical University of South Carolina for sphingolipid analysis. We thank the BCM Intellectual and Developmental Disabilities Research Center (IDDRC) confocal microscopy core, which is supported by the Eunice Kennedy Shriver National Institute of Child Health and Human Development (1U54 HD083092). We acknowledge support from the Friedreich's Ataxia Research Alliance, NIH (1RC4GM096355), the Robert A and Renee E Belfer Family Foundation, the Huffington Foundation, and Target ALS to HJB. HJB is an Investigator of the Howard Hughes Medical Institute.

## Additional information

**Competing interests**

HJB: Reviewing editor, *eLife*. The other authors declare that no competing interests exist.

## Funding

| Funder | Grant reference number | Author |
| --- | --- | --- |
| Friedreich's Ataxia Research Alliance | | Kuchuan Chen Guang Lin Hugo J Bellen |
| Robert A. and Renee E. Belfer Family Foundation | | Hugo J Bellen |
| Howard Hughes Medical Institute | | Hugo J Bellen |
| National Institutes of Health | 1RC4GM096355 | Hugo J Bellen |
| The Huffington Foundation | | Hugo J Bellen |
| Target ALS | | Hugo J Bellen |

The funders had no role in study design, data collection and interpretation, or the decision to submit the work for publication.

## Author contributions

KC, TS-YH, GL, KLT, Conception and design, Acquisition of data, Analysis and interpretation of data, Drafting or revising the article; MNR, HJB, Conception and design, Analysis and interpretation of data, Drafting or revising the article

## Author ORCIDs

Matthew N Rasband, http://orcid.org/0000-0001-8184-2477
Hugo J Bellen, http://orcid.org/0000-0001-5992-5989

## Ethics

Human subjects: Postmortem cardiac tissue was provided by Dr. Arnulf Koeppen and collected at Veterans Affairs (VA) Medical Center, Albany, NY, USA. Our receipt and analyses of these postmortem tissues was conducted with the approval of the Institutional Review Board at Baylor College of Medicine (protocol H-32191).

Animal experimentation: All mouse work and procedures were approved by the Animal Care and Use Committee at Baylor College of Medicine and were performed in accordance with the guidelines of the U.S. National Institutes of Health (animal protocol: an-4634).

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
