## [Decision Letter]

Thank you for submitting your article "Loss of *Frataxin* activates the iron/sphingolipid/PDK1/Mef2 pathway in mammals" for consideration by *eLife*. Your article has been favorably evaluated by a Senior Editor and three reviewers, one of whom is a member of our Board of Reviewing Editors. The following individual involved in review of your submission has agreed to reveal his identity: Bingwei Lu (Reviewer #2).

The reviewers have discussed the reviews with one another and the Reviewing Editor has drafted this decision to help you prepare a revised submission.

Summary:

This is a short manuscript submitted as a Research Advance that builds upon a recent publication in *eLife*. The earlier *eLife* paper characterized the phenotype caused by loss of function of *Drosophila Frataxin*. The fly mutant showed iron accumulation in the nervous system and activation of a pathway that involves Pdk1 and Mef2 that triggered degeneration, but contrary to expectations, no evidence of ROS. In the current paper, the authors build upon this story by examining the consequence of FXN loss of function in young mice. They validate a sgRNA from mouse FXN in vitro, then use AAV-mediated intraventricular delivery to P0 Rosa26-Cas9 mice, and this results in knockdown of endogenous FXN as assessed by mRNA levels. At P130 the knockdown mice are small, kyphotic, and perform poorly on hanging wire and rotarod. Histologically, the cortices show upregulation of ATF3 and loss of AnkG, as well as deposition of Fe^2+^ and Fe^3+^. No elevation of 4-HNE was detected, suggesting no major oxidative stress. Elevated levels of phosphorylated PDK1 and modestly elevated levels of several mRNAs purported to be downstream of Mef2 were detected. Sphingolipid levels in the mice were not examined. Examination of postmortem heart tissue from several FRDA patients showed increased sphingolipid levels and increased levels of phosphorylated PDK1.

As may be surmised from the longer-than-usual review period, the editors and reviewers engaged in considerable discussion regarding your submission. This included not only discussion of the data in your submission bit also the definition of an *eLife* Research Advance. We feel that this new work adds significantly to the message of the original paper. Nevertheless, the data in the research advance still needs to be as solid and persuasive as any other data published in *eLife*. For this manuscript we believe there are 4 concerns that need to be addressed:

1) Is the methodology used to detect Fe accumulation sound? As the authors' point out, there is a discrepancy between the identification of Fe^3+^ deposition in this paper and the absence of this finding in a prior conditional knockout (Puccio et al.). The authors suggest that their assay is more sensitive because they amplify the signal with DAB. Is there precedent for this or can they validate this purported increased sensitivity? Can they show this works in tissue from one of the other models to corroborate?

2) Are there regional vulnerabilities to FXN loss within the brain of knockdown mice? In Friedreich's ataxia the loss of FXN most profoundly impacts dorsal root ganglia, spinocerebellar tracts, lateral corticospinal tracts, and posterior columns. The current paper examines only the cortex. Why?

3) A principal conclusion of the earlier paper is that FXN loss resulted in iron accumulation and consequential induction of sphingolipid synthesis, and that this pathway is relevant to the observed degeneration. These FXN knockdown mice exhibit degeneration, but whether this is preceded by increased sphingolipid synthesis is not addressed.

4) The conclusion that the iron/ sphingolipid/ PDK1/Mef2 pathway is operative in this degeneration would be strengthened by examined with cellular resolution by histology, and there was evidence of iron accumulation in the same neurons that show reduced FXN expression (or at least GFP signal), and that these neurons show activation of PDK1/Mef2.

Some of these questions may be addressed in the text, but we anticipate that others will require additional experiments. Since these additional experiments are largely histological in nature, we expect that they should not take long, assuming the lab has tissue available. If a few more weeks than the official guideline of 2 months is required please let us know.

---

## [Author Response]

*[…] As may be surmised from the longer-than-usual review period, the editors and reviewers engaged in considerable discussion regarding your submission. This included not only discussion of the data in your submission bit also the definition of an eLife Research Advance. We feel that this new work adds significantly to the message of the original paper. Nevertheless, the data in the research advance still needs to be as solid and persuasive as any other data published in eLife. For this manuscript we believe there are 4 concerns that need to be addressed:*

*1) Is the methodology used to detect Fe accumulation sound? As the authors' point out, there is a discrepancy between the identification of Fe^3+^ deposition in this paper and the absence of this finding in a prior conditional knockout (Puccio et al.). The authors suggest that their assay is more sensitive because they amplify the signal with DAB. Is there precedent for this or can they validate this purported increased sensitivity? Can they show this works in tissue from one of the other models to corroborate?*

Using Perls’ staining with DAB enhancement is a classical protocol to detect ferric irons, and the protocol was first described in 1980 (Nguyen-Legros et al., 1980). The method has been used in several animal models and tissues, including fly (Kosmidis et al., 2012), zebrafish (Lumsden et al., 2007), and mouse (van Duijn et al., 2013). We have validated the sensitivity by comparing the signals with either Perls’ staining alone or Perls’/DAB staining (Figure 3 and Figure 3—figure supplement 2), and we only observed an obvious increase in signal in the brain sections by using Perls’/DAB staining. The data clearly show that the Perls’/DAB staining is a more sensitive method to detect ferric irons. We described this method in our previous paper as well and show its application in third instar larval brains. Finally, we were not able to obtain the published *Fxn* conditional knock-out mice produced by Helen Puccio and colleagues after 5 requests and several phone calls. Therefore, we cannot test Perls’/DAB staining in that mouse model.

*2) Are there regional vulnerabilities to FXN loss within the brain of knockdown mice? In Friedreich's ataxia the loss of FXN most profoundly impacts dorsal root ganglia, spinocerebellar tracts, lateral corticospinal tracts, and posterior columns. The current paper examines only the cortex. Why?*

The primary affected areas in Friedrich’s ataxia patients are the dorsal root ganglia, the dentate nucleus, and the areas mentioned above. Since we are not able to obtain the *Fxn* conditional knock out mouse, we resorted to injections of the *Fxn*-sgRNA cloned in AAV into the Rosa26-Cas9 neonatal animals (Kim et al., 2014). The virus spreads into the brain and infected neurons efficiently in cerebral cortex but poorly in the cerebellum and the virus did not reach the dorsal root ganglia. Hence, we choose the cerebral cortex for phenotypic analysis.

*3) A principal conclusion of the earlier paper is that FXN loss resulted in iron accumulation and consequential induction of sphingolipid synthesis, and that this pathway is relevant to the observed degeneration. These FXN knockdown mice exhibit degeneration, but whether this is preceded by increased sphingolipid synthesis is not addressed.*

Since iron, PDK1, and Mef2 activities are obviously increased in mice at P60 (Figure 3), and based on the yeast and fly data, it seems very likely that the iron/sphingolipid/PDK1/Mef2 pathway is activated at P60, before the mice exhibit behavioral defects at P130. In addition, the patient heart samples show increased sphingolipids, further supporting our hypothesis. We recently obtained sphingolipid analyses of single *LacZ*-sgRNA and *Fxn*-sgRNA mice at P60 after waiting for many months for the results. The levels of several sphingolipid species are indeed increased, consistent with our other data (see Figure 5). We did not integrate these data as we feel we need at least 3 samples for a proper analysis. Unfortunately, we currently do not have any mouse brain samples as they were used for all other assays and histochemical stainings. We can re-inject the virus and perform sphingolipid analysis, but it will take at least 5 months to get the data (3 months for mice and 2-3 month for the sphingolipid mass spectrometry). However, given that we already published the pathway in flies and that the hearts of patients show this up-regulation, we feel that we have compelling evidence that this pathway is evolutionarily conserved.

If the reviewers feel that we should include the data, we are willing to integrate the sphingolipid lipid data at P60 in the manuscript as supplemental data with the caveat that they are only based on a single animal.

Author response image 1.Sphingolipid levels of *LacZ*- and *Fxn*-sgRNA mice.**DOI:**
http://dx.doi.org/10.7554/eLife.20732.012

4) The conclusion that the iron/ sphingolipid/ PDK1/Mef2 pathway is operative in this degeneration would be strengthened by examined with cellular resolution by histology, and there was evidence of iron accumulation in the same neurons that show reduced FXN expression (or at least GFP signal), and that these neurons show activation of PDK1/Mef2.

We agree that examining the phenotype at the single cell resolution will provide us with more and better information about the mechanism. Unfortunately, there is no anti-FXN antibody nor are there anti- phosphorylated PDK1 antibodies available for immunostaining to detect the endogenous FXN levels or PDK1 phosphorylation levels. The anti-FXN antibody produced by Helen Puccio was not made available upon repeated requests and the available commercial antibodies only work for immunoblots. There are also no reagents available to examine Mef2 activity by immunostaining. However, we performed experiments with some formerly made brain sections and examined the iron distribution in the *Fxn*- sgRNA mice. We found that the DAB signal only co-localized with a neuronal marker and not with glial markers (for astrocytes and microglia), suggesting that iron mostly accumulates in neuronal cells. These data have now been incorporated (Figure 3—figure supplement 1).